# Descriptive Sensory Analysis of Pizza Cheese Made from Mozzarella and Semi-Ripened Cheddar Cheese Under Microwave and Conventional Cooking

**DOI:** 10.3390/foods9020214

**Published:** 2020-02-19

**Authors:** Nabila Gulzar, Aysha Sameen, Rana Muhammad Aadil, Amna Sahar, Saima Rafiq, Nuzhat Huma, Muhammad Nadeem, Rizwan Arshad, Iqra Muqadas Saleem

**Affiliations:** 1Department of Dairy Technology, University of Veterinary and Animal Sciences, Lahore 55300, Pakistan; nabila.gulzar@uvas.edu.pk (N.G.); muhammad.nadeem@uvas.edu.pk (M.N.); iqra.muqadas@uvas.edu.pk (I.M.S.); 2National Institute of Food Science & Technology, University of Agriculture Faisalabad, Faisalabad 38000, Pakistan; dilrana89@gmail.com (R.M.A.); drnuzhathuma@gmail.com (N.H.); 3Department of Food Science and Technology, University of Poonch, Rawalakot 12350, Pakistan; Saimaft2009@gmail.com; 4University Institute of Diet and Nutritional Sciences, The University of Lahore, Gujrat Campus, Gujrat 50700, Pakistan; rizwanarshad1129@gmail.com

**Keywords:** appearance, cheese, descriptive, flavor, sensory

## Abstract

The present study used descriptive sensory analysis (DSA) to compare Pizza cheeses prepared from various combinations of fresh Mozzarella and semi-ripened Cheddar cheeses and cooked under conventional and microwave cooking methods. A cheese sensory lexicon was developed, and descriptive sensory profiles of the Pizza cheeses were evaluated using a panel of semi-trained judges (*n* = 12). The following characteristics, flavor (cheddar, acidic, rancid, bitter, salty, creamy, and moldy), texture (stringiness, stretchability, firmness, and tooth pull), and appearance (meltability, oiliness, edge browning, and surface rupture) of Pizza cheeses were analyzed and compared with control samples. The sensory analysis of Pizza cheeses showed more preference toward a higher level of ripened Cheddar cheese (4 months), which was cooked using the microwave. However, the scores for texture properties were decreased with the addition of the semi-ripened cheese. The scores for stretchability and tooth pull were high in the microwave cooked samples compared with the conventionally cooked samples. The appearance attributes (meltability, oiliness, and edge browning) scores were increased with the increasing of ripened Cheddar cheese content while surface rupture was decreased. Microwave cooked Pizza cheese showed better meltability and oiliness but lower edge browning scores. The results showed that amalgamations of fresh Mozzarella and semi-ripened Cheddar cheese had a significant (*p* < 0.05) and positive effects on the sensory qualities of Pizza cheeses.

## 1. Introduction

Cheese is the most diverse group of dairy products available in a wide range of flavors and forms. Processed cheese is a type of cheese prepared from natural cheese that is mixed with other ingredients [1]. Sensory quality has been recognized as a crucial aspect of sale and marketing of cheese. In the dairy industry, traditional methods are used for the evaluation of sensory quality but nowadays new methods have been developed for the sensory analysis [2]. Various descriptive sensory methods such as texture profile method [3], flavor profile method [4], quantitative descriptive analysis ^TM^ [5], the Spectrum ^TM^ Method [6], free choice profiling [7], quantitative flavor profiling [8], and generic descriptive analysis have been reported in literature. These different approaches with the exception of the generic descriptive analysis use a combination of descriptive analyses to meet specific objectives.

Descriptive sensory analysis is considered the most powerful sensory tool in cheese research. In this technique, a panel of 6–12 individuals are trained to identify and quantify the sensory aspects of food, which may include appearance, aroma, flavor, texture, or any single aspect [9]. Descriptive sensory analysis is distinguished from other sensory testing methods in that it seeks to profile all perceived sensory characteristics of a product [10].

Fresh Mozzarella cheese shows a fibrous structure and melting and stretching characteristics upon cooking and use on pizza but lacks pleasant flavor characteristics. Similarly, ripened cheese lacks desirable texture but has good flavor characteristics [8,11]. Cheddar cheese undergoes a complex series of chemical, bacterial, and enzymatic reactions during ripening [2,12], which are responsible for the development of the characteristic sensory profile that is typical of ripened cheese [13,14,15]. Almost 40% of the pizza restaurants are using cheese blends to create different flavors from that of their competitors. Therefore, the blending of cheeses has been used to develop desirable sensory characteristics that support marketing of pizza.

Microwave heating has many advantages over conventional heating. In microwave heating, the rate of heat transfer is greater while organoleptic properties, color, the original flavor of food, and nutritional value are maintained [16]. Microwave-circulation water combination heating has been used to heat macaroni and cheese due to its rapid and uniform heat transfer. Consumers liked macaroni and cheese that was treated with microwave more than freshly cooked [17]. However, the effect of microwave heating on the quality characteristics of Pizza cheese such as flavor, texture, and color has not been investigated. The aim of the present study is to investigate the sensory characteristics (texture, flavor, and appearance) of Pizza cheeses prepared from Mozzarella and semi-ripened Cheddar cheese combinations and cooked in microwave and conventional oven using a Descriptive Sensory Analysis technique.

## 2. Materials and Methods

Raw buffalo milk for the preparation of Mozzarella Cheese (MC) and Cheddar Cheese (CC) was obtained from the SB Dairy Farm Jhapal, Faisalabad, Pakistan. Mesophillic and thermophillic starter cultures (Chr. Hansen Ireland Ltd., Rohan Industrial Estate, Little Island, Co. Cork, Ireland) were used for the making of Cheddar and Mozzarella cheese, respectively. Chymosin (50,000 u/G strength of Top Pharm Chemical Co., Ltd. Shanxi, China) was used as coagulant in both cheeses.

### 2.1. Preparation of Mozzarella, Cheddar, and Pizza Cheeses

Cheddar and Mozzarella cheeses were prepared by the methods of Ong et al. [18] and Zisu and Shah [19], respectively. Pizza cheeses were manufactured by blending different proportions of Mozzarella and Cheddar cheeses, water (5%), and emulsifying salts (2% Tri-sodium citrate) in a steam jacketed cooker (Blentech Corp., Rohnert Park, CA, USA) at 80 °C for 10 min. Seven Pizza cheese sample groups were prepared; PC_0_ group with 100% Mozzarella cheese was used as control. Three sample groups (PC_1_, PC_2_, and PC_3_) were manufactured with the amalgamation of (75:25), (50:50), (25:75) of Mozzarella cheese and semi-ripened Cheddar cheese (2 months) whereas, other three groups (PC_4_, PC_5_, and PC_6_) were made with (75:25), (50:50), (25:75) of Mozzarella cheese and semi ripened Cheddar cheese (4 months), respectively [20]. The molten cheese was molded in pans and cooled overnight at 5 °C. Pizza cheese blocks were then repacked, vacuum-sealed, and stored again at 5 °C for further analysis.

### 2.2. Physicochemical Analysis of Mozzarella, Cheddar, and Pizza Cheeses

Chemical analysis of moisture [21], protein [22], fat [23], total calcium, and water-soluble calcium [24] of natural and Pizza cheeses were determined by following standard methods.

### 2.3. Descriptive Sensory Analysis of Pizza Cheeses

#### 2.3.1. Panel Selection and Training

To carry out the descriptive sensory analysis, 12 judges (faculty members) were recruited from the National Institute of Food Science and Technology, University of Agriculture, Faisalabad, based on their availability and willingness to participate in this study. The panelists were screened and trained initially based on their sensitivity to recognize basic tastes [25] and ranking tests as well as capacity to identify discrimination by a triangular test using Pizza cheese [26]. Five training sessions were conducted to define sensory terminologies before the final examination. Test samples were served as descriptive stimuli for the development of language during the training session. Sensory descriptors for flavor (cheddary, acidic, rancid, bitter, salty, sweet, creamy, and moldy), texture (stringiness, stretchability, firmness, tooth pull) and appearance (meltability, oiliness, edge browning, and surface rupture) of Pizza cheeses were developed [27]. The sensory descriptors along with their definitions and evaluation techniques are given in Table 1**.**

#### 2.3.2. Sensory Profiling of Pizza Cheeses

Sensory profiling of Pizza cheeses subjected to two cooking methods (microwave and conventional as described below) was performed by descriptive sensory analysis. For sensory analysis, 50 g of each shredded Pizza cheese sample was topped on pizza dough for both cooking methods. The microwave heat treatment consisted of heating Pizza cheese for 2 min in a microwave oven at 900 W power (Dawlance Model DW-395HP, Lahore, Pakistan). Cooking in a conventional oven was carried out for 15 min at 177 °C in Sears Kenmore Model 73791 gas oven (Chicago, IL, USA). The Pizza cheese samples after cooking were presented to the panelists just after being removed from the oven in a sensory evaluation laboratory at the National Institute of Food Science and Technology, University of Agriculture, Faisalabad Pakistan. Panelists scored the samples for flavor, texture, and appearance descriptors of Pizza cheeses by measuring sensory intensities autonomously in an individual sensory booth without any intensity standard references. The sensory intensities were evaluated by using a line scale from 0 to 10 from left to right; where 0 means the sensory attribute was not detected and 10 means the attribute is present at a high concentration. Rating of the sensory attributes was performed within a scale of 1–10. Each sensory evaluator assessed each Pizza cheese sample three times and descriptive scores were given to various descriptors within a total of 10 [27].

#### 2.3.3. Statistical Analysis

The difference between products was identified by one-way analysis of variance. The means were compared using Tukey’s test after their significant difference at a 5% level of significance. The data were presented as mean values ± standard error (SE). Principal component analysis and cluster analysis were used for the dataset in the descriptive sensory analysis [28].

## 3. Results and Discussions

### 3.1. Physicochemical Composition of Cheeses

The mean values for the physicochemical composition of Mozzarella, Cheddar, and Pizza cheeses are depicted in Table 2. The results indicated that moisture content was decreased while protein, fat, and calcium were increased in Pizza cheeses as compared to control (100% Mozzarella). This variation in composition was due to different manufacturing procedures, starter culture, and ripening period of both kinds of cheese. However, water-soluble calcium was decreased in Pizza cheeses, which might be due to differences in milling, pH, and ripening period of both kinds of cheese.

### 3.2. Descriptive Sensory Evaluation of Pizza Cheese

#### 3.2.1. Flavor Descriptors of Pizza Cheese

The sensory scores of flavor descriptors of Pizza cheeses are shown in Table 3. It can be observed that more appealing flavors were obtained for Pizza cheeses that contained 4 months ripened Cheddar cheese compared to the 2 months ripened Cheddar cheese. Furthermore, higher scores were obtained as the level of Cheddar cheese was increased in the Pizza cheese.

It has been observed that amalgamation of Mozzarella and ripened Cheddar cheeses is a useful process to enhance the flavor characteristics of Pizza cheese. Biochemical changes that occur in ripened Cheddar are responsible for the enhanced flavor of Pizza cheese [2]. More ripened Cheddar cheese imparts a rich flavor to the Pizza cheese, which is related to higher fat and protein contents, and glycolysis, lipolysis, and proteolysis processes occurring during maturation [29]. The lipolytic and oxidative changes in Cheddar cheese also contribute specific flavor and enhance sensory acceptability of Pizza cheese. For example, free fatty acids contribute to flavor precursors to cheese such as acids from C4:0 and C12:0, which impart specific flavors i.e., rancid, sharp, goaty, soapy, and coconut-like [30]. In Pizza cheese, the proportion of hydrolytic products of protein (peptides and amino acids) was increased with the level and age of Cheddar cheese that may also be responsible for the development of suitable cheese flavor. The acidic flavor in Pizza cheese is due to lactate that is generated in many reactions such as oxidation and microbial metabolism [31].

Cheese baked in microwave oven samples had higher sensory scores than conventionally baked samples for cheddary, acidic, sweet, creamy, and moldy flavor (Table 3). The highest score of bitterness flavor was found in PC_6_-M followed by PC_6_-C. There were no differences in rancidity among PC_5_-M and PC_5_-C, PC_4_-M, and PC_4_-C treatment groups. The flavor characteristics of Pizza cheese in microwave cooked Pizza are due to the short and uniform heating of the upper surface of pizza compared to the conventional cooking that encountered a long processing time that caused losses in product flavor as well as development of new cooked cheese notes [32].

#### 3.2.2. Texture Descriptors of Pizza Cheese

Intensity responses to texture descriptors of Pizza cheeses with respect to ripening time (months), proportion of cheese amalgamation, and cooking methods are depicted in Table 4. The sensory response revealed that the lowest stringiness scores were awarded to the PC_0_ (control) group while the highest mean scores were given to the PC_6_ treatment group which had 25% Mozzarella cheese and 75% 4 months ripened Cheddar cheese (Table 4). For all other textural attributes i.e., stretchability, firmness, tooth pull, the highest scores were given to control and the lowest score values were found in PC_6_. The results indicate that ripening months and level of cheese influenced the texture characteristics of Pizza cheese.

Stretchability, firmness, and tooth pull changes may be attributed to differences in manufacturing and level of intact casein of both kinds of cheese [33]. The reduction in stretchability, firmness, and tooth pull of all Pizza cheese with ripening time (months) and increasing level of Cheddar cheese is associated with a reduction in the intact casein [33], a high degree of casein hydration, and low Ca content [34]. These conditions reduce cross-linkages and casein intermolecular associations of network. Therefore, the texture attributes decrease with increasing level and age of Cheddar cheese [35].

The amalgamation of cheese improved stringiness due to better emulsification of fat in amalgamated Pizza cheese that evenly surrounds the protein network and form channels during heating of cheese. These channels allow the formation of protein fibers and separates them and consequently gives the Pizza cheese better stringiness characteristics. When Mozzarella cheese is only used on Pizza, more protein interactions, and less fat emulsification, a fusion of the protein matrix takes place that reduces formation of strands occurring during heating [34].

The appreciably higher scores for meltability and oiliness of amalgamated Pizza cheese than control cheese are due to the result of substantial difference in protein hydration in both kinds of cheese, as the control Pizza cheese contains a lower level of protein hydration while in amalgamated Pizza cheese hydration level increased due to Cheddar cheese [33]. The score for browning increased due to the amalgamation, which might be due to the difference in hydrolyzed sugars in cheese, since the Cheddar cheese has more monosaccharides than Mozzarella due to breakdown of residual lactose [36]. Surface rupture was higher in the control samples, owing to the heterogeneous structure of Mozzarella cheese while in the amalgamated Pizza cheese, the homogenous structure and emulsification of fat reduces surface rupture [37]. An amalgamation of cheeses of different ripening periods is an important strategy to achieve the desired structure development of cheese. In contrast to this study, Lenze et al. [38] adopt rework methodology of upstream homogenization for the structure formation in processed cheese [34]. However, in both studies weak physical bonds were responsible for the structure development of cheese.

The texture characteristics were also considerably influenced by the pizza cooking methods. The average scores of Pizza cheese for tooth pull and stretchability attained using microwave oven were higher than a conventional oven. The sensory scores of Pizza cheese for firmness and stringiness were lower in the microwave cooked samples compared to Pizza cheese cooked using the conventional oven.

The improvement in stretching and tooth pull associated with the decrease in firmness and stringiness with microwave cooking is related to the steam production in a microwave. This steam produces driving pressure that leads to the expansion and softness of cheese matrix [39]. Although the texture characteristics of Pizza cheese were improved by microwave heating, other quality defects arise such as soggy texture of the pizza’s dough, which is due to short heat processing time and less penetration power [40]. This problem can be solved by using an oven that has both a microwave and conventional oven heating system.

#### 3.2.3. Appearance Descriptors of Pizza Cheese

Regarding appearance, descriptors of Pizza cheese of lower meltability, oiliness, and edge browning scores were found in control Pizza cheese (PC_0_) while these characteristics were improved with the amalgamation of Cheddar cheese in Mozzarella cheese. The highest scores were awarded to Pizza cheese (PC_6_) prepared with 25% Mozzarella and 75% level of 4 months ripened Cheddar cheese (Table 5). However, the surface rupture trait decreased with the amalgamation of cheese, with the lowest and the highest mean scores were awarded to PC_6_ and PC_0_, respectively.

The meltability and oiliness attributes of Pizza cheese from PC_0_–PC_6_ were increased and this could be explained by the changes in the microbiological, biochemical, and metabolic properties of the cheeses [12,41], most importantly the proteolysis process in the Cheddar cheese. The abovementioned changes could reduce the protein–protein interactions and fat globules coalescence, which are primarily present in a dispersed form in the protein matrix. Collectively, this could lead to increased meltability and oiliness upon heating the Pizza cheese [37,42]. The increased melting and oiliness in Pizza cheeses containing the 4 months ripened Cheddar cheese had more casein hydration which changes the state of water and protein within the cheese matrix [43]. The lipolysis, proteolysis, and hydrolysis of lactose as well as increased concentration of galactose, all may cause browning due to reactivity and interactions leading to formation of Millard reaction products [44]. Lee et al. [36] suggested similar results for Cheddar cheese which is more susceptible to browning due to higher concentrations of reaction substrates produced during ripening. The reduced surface rupture may be associated with increased emulsification of fat and homogeneous structure of Pizza cheese [36,37].

The change in the appearance of Pizza cheese during heating was influenced by the cooking methods. The mean scores for edge browning and surface rupture of microwave cooked Pizza cheese was comparatively lower than conventional oven-cooked Pizza cheese.

In the microwave-cooked Pizza cheese, meltability and oiliness scores were higher while surface rupture was lower than the conventionally cooked pizza. This may be due to fast migration of moisture that allows rapid flow of oil to the surface of Pizza cheese and reduced surface rupture. Unlike conventional oven-cooked Pizza cheese, microwave oven cooked showed less edge browning, and could be due to the reason that a microwave does not attain the desired temperature that causes browning or caramelization [45].

### 3.3. Multivariate Analysis of Sensory Scores

Principal component analysis (PCA) showing the first two principal components (PCs) of descriptive sensory analysis of Pizza cheese is shown in Table 6. PC1 and PC2 significantly (*p* < 0.05) discriminated between the cheeses and accounted for 78.8% and 10.9% of the variation, respectively. Figure 1 shows a bi-plot of the loadings scores of the attributes. Hierarchical cluster analysis (HCA) of the raw data was used to cluster closely related Pizza cheeses in terms of sensory characters and generated four main clusters (Figure 2).

PC1 explains the variation (78.8%) between the sensory characters of the Pizza cheeses separated on the basis of amalgamation of cheeses (Mozzarella and Cheddar) at different levels and ripening months of Cheddar cheese. The amalgamation of cheeses at various levels influenced the sensory perception of Pizza cheese regardless of ripening time (months) of the Cheddar cheese. PC2, which accounted for 10.9% of the variation between the Pizza cheeses, distinguished the samples on the basis of cooking methods used. The relative positions of Pizza cheeses on the bi-plot is a useful index of the impact of amalgamation of cheeses (Mozzarella and Cheddar), their levels, and ripening time of the Cheddar cheese on the sensory profile of Pizza cheeses.

HCA offers a basis for the interpretation of the bi-plot and the identification of clusters of closely related samples. Grouping and subgrouping on the dendrogram (Figure 2) indicated that amalgamation of cheeses, their levels, and ripening time of the Cheddar cheese significantly (*p* < 0.05) influenced the sensory characteristics and all took part in laying the foundations of clusters. Similarly, the positions of Pizza cheese on the PCA plot also showed a significant effect for these parameters on the sensory perception of Pizza cheeses.

## 4. Conclusions

Considering the sensory evaluation, it is concluded that the flavor and texture of Pizza cheeses with higher proportions of 4 months ripened Cheddar cheese were liked more by panelists. While, appearance of Pizza cheese with 50:50% combination of Mozzarella and Cheddar with 4 months ripened attained higher scores. Microwave oven cooking had better flavor development, while conventional cooking resulted in better texture properties. Pizza cheese with 2 months ripened Cheddar cheese showed better textural attributes and stretchability, while Pizza cheese with 4 month ripened Cheddar rated well for flavor, meltability, and appearance. Therefore, it was very difficult to select one combination that fulfilled all the requirements of optimized Pizza cheese. However, the most recommended ratio for Pizza cheese was 75% Mozzarella and 25% 2 months ripened Cheddar cheese as compared to other combinations.

## Figures and Tables

**Figure 1 foods-09-00214-f001:**
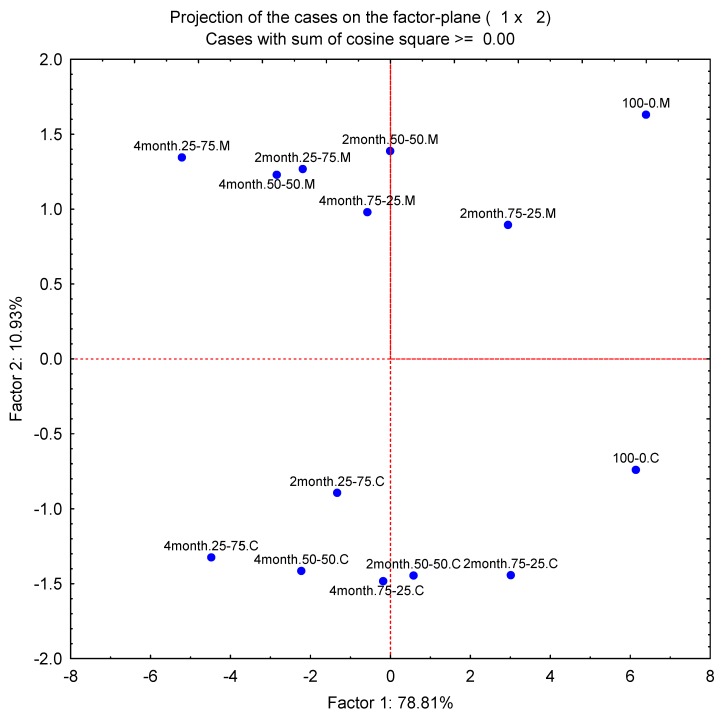
Principal component analysis (PCA) showing the first two principal components (PCs) of descriptive sensory analysis of Pizza cheese. M = Microwave cooking, C = Conventional oven cooking, 2 and 4 months = 2 and 4 months ripened Cheddar cheese, 75:25, 50:50, 25:75 = the first part represents Mozzarella and the second part represents Cheddar cheese proportion in Pizza cheeses.

**Figure 2 foods-09-00214-f002:**
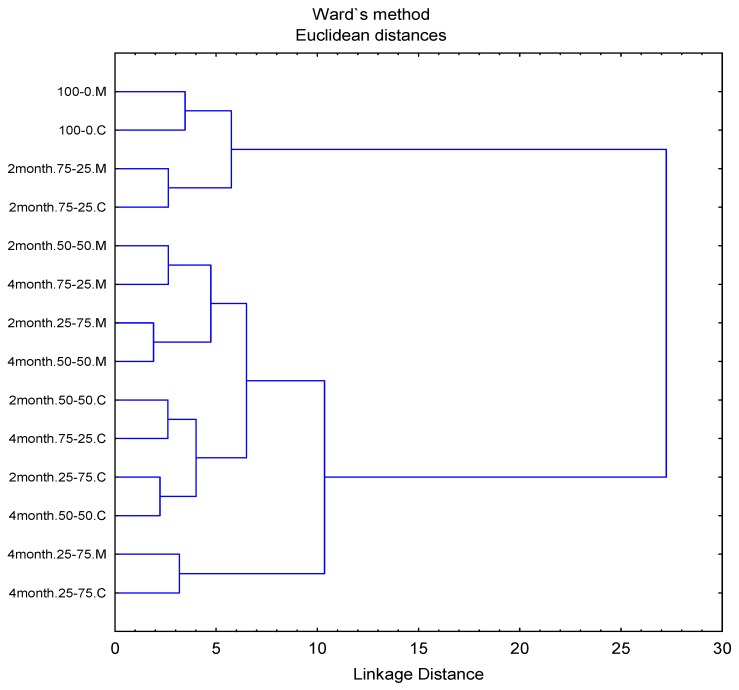
Dendrogram obtained from hierarchical cluster analysis (HCA) of sensory data of Pizza cheeses. M = Microwave cooking, C = Conventional oven cooking, 2 and 4 months = 2 and 4 months ripened Cheddar cheese, 75:25, 50:50, 25:75 = the first part represents Mozzarella and the second part represents Cheddar cheese proportion in Pizza cheeses.

**Table 1 foods-09-00214-t001:** Sensory descriptors and definitions along with their evaluation technique.

**Flavor Descriptors**		
**Cheddar flavor**	Intensity of cheese flavor	Take a specific amount of cheese in mouth and observe the intensity of flavor attributes with the sense of taste and smell
**Acidic**	Intensity of lactic acid	
**Rancid**	Intensity of soapy flavor	
**Bitter**	Intensity of caffeine	
**Salty**	Intensity of salt	
**Sweet Creamy Moldy**	Intensity of sugar taste. Intensity of strong but bland flavor. Intensity of pungent flavor	
**Texture Descriptors**		
**Stringiness**	Number of strings that result when a fork is put into the cup and the cheese is pulled up	Dip fork into center of sample and extend vertically, simultaneously evaluating stringiness and stretchiness
**Stretchability**	Length of the strings that result when a fork is put into the cup and cheese is pulled up	
**Firmness**	Force required to compress between tongue and palate	Place 3 g of sample between tongue and press against palate to evaluate firmness 1, then move sample between molars, bite down, and evaluate tooth pull
**Tooth pull**	Force required for the teeth to separate from the sample	
**Appearance Descriptors**		
**Descriptors**	**Definition**	**Evaluation Techniques**
**Meltability**	Homogeneity of the sample	Look at the top view of the sample on the pizza topping and determine quantity of descriptor present
**Oiliness**	Quantity of oil resting on the top of the sample	
**Edge browning**	Amount of browning present on the edges of the sample	
**Surface rupture**	Diversity of the surface of the sample	

**Table 2 foods-09-00214-t002:** Chemical composition of Mozzarella, Cheddar, and Pizza cheeses.

Components	Mozzarella Cheese	Cheddar Cheese		Pizza Cheeses						
		2 Months Ripened	4 Months Ripened	PC_0_ 100:0 Mo:Ch	PC_1_ 75:25 Mo:Ch.2	PC_2_ 50:50 M:Ch.	PC_3_ 25:75 M:Ch.2	PC_4_ 75:25 Mo:Ch.4	PC_5_ 50:50 Mo:Ch.4	PC_6_ 25:75 Mo:Ch.4
**Moisture %**	46.95 ± 1.00	35.36 ± 47	35.26 ± 0.20	51.23 ± 0.87A	50.80 ± 0.62 B	50.00 ± 0.90 C	47.54 ± 0.57D	49.72 ± 1.00 C	47.81 ± 0.57 D	46.58 ± 0.33 E
**Protein %**	25.36 ± 1.18	28.23 ± 0.20	28.20 ± 0.10	25.37 ± 0.54B	26.07 ± 0.28B	26.70 ± 0.49AB	27.51 ± 0.208A	26.04 ± 0.52B	26.40 ± 0.57AB	27.58 ± 0.43A
**Fat %**	23.25 ± 1.80	29.33 ± 1.15	29.03 ± 1.19	23.00 ± 1.00B	24.32 ± 0.66B	26.67 ± 0.33A	27.64 ± 0.66A	24.33 ± 0.33B	27.32 ± 0.88A	28.35 ± 0.88A
**Total Calcium mg/100g**	650 ± 16.21	754 ± 15.92	749 ± 15.83	652 ± 26D	676 ± 22CD	702 ± 24ABC	728 ± 28AB	689 ± 26BCD	722 ± 29ABC	748 ± 30A
**Water Soluble Calcium mg/100g**	310 ± 15.32	220 ± 7.22	240 ± 6.21	313 ± 12.5A	287 ± 9.0B	265 ± 10.5C	242 ± 8.0D	309 ± 10.0A	285 ± 12.0B	262 ± 8.5C

Means within a row with different superscripts differ significantly (*p* < 0.05), Mo = Mozzarella cheese, Ch. = Cheddar cheese; Ch.2 = 2 months ripened Cheddar cheese, Ch.4 = 4 months ripened Cheddar cheese; values given are the means of the three replicates.

**Table 3 foods-09-00214-t003:** Effect of cheese proportions, ripening time (months), of Cheddar cheese and cooking methods on the flavor descriptors of Pizza cheese.

Treatments	Cheese Proportions	Flavor Descriptors (Mean ± SE)							
		CHEDDAR	ACIDIC	RANCID	BITTER	SALTY	SWEET	CREAMY	MOLDY
**PC_0__M**	**Mo:Ch.** **100:0**	4.86 ± 0.37 ^g^	4.00 ± 0.22 ^h^	2.00 ± 0.22 ^ef^	1.43 ± 0.20 ^f^	2.71 ± 0.29 ^c^	2.37 ± 0.37 ^g^	1.97 ± 0.32 ^g^	2.95 ± 0.32 ^g^
**PC_0__C**		3.57 ± 0.34 ^def^	3.71 ± 0.18 ^h^	1.71 ± 0.18 ^f^	1.29 ± 0.18 ^f^	2.86 ± 0.34 ^bc^	1.76 ± 0.34 ^def^	1.76 ± 0.24 ^def^	1.56 ± 0.24 ^def^
**PC_1__M**	**Mo:Ch.2** **75:25**	4.71 ± 0.26 ^fg^	5.71 ± 0.18 ^def^	2.71 ± 0.29 ^cd^	2.57 ± 0.20 ^e^	3.43 ± 0.20 ^abc^	3.76 ± 0.25 ^fg^	3.36 ± 0.35 ^fg^	3.36 ± 0.35 ^fg^
**PC_1__C**		4.14 ± 0.29 ^ef^	4.43 ± 0.30 ^gh^	2.57 ± 0.30 ^de^	2.43 ± 0.30 ^e^	3.57 ± 0.20 ^ab^	2.15 ± 0.29 ^ef^	2.25 ± 0.29 ^ef^	2.25 ± 0.29 ^ef^
**PC_2__M**	**Mo:Ch.2** **50:50**	5.43 ± 0.26 ^de^	6.00 ± 0.31 ^cde^	3.43 ± 0.20 ^ab^	2.71 ± 0.18 ^e^	3.43 ± 0.20 ^abc^	4.79 ± 0.21 ^de^	5.19 ± 0.11 ^de^	4.19 ± 0.11 ^de^
**PC_2__C**		5.14 ± 0.30 ^cde^	5.00 ± 0.22 ^fg^	3.29 ± 0.29 ^abc^	2.57 ± 0.20 ^e^	3.71 ± 0.18 ^a^	4.10 ± 0.30 ^cde^	4.90 ± 0.20 ^cde^	3.90 ± 0.20 ^cde^
**PC_3__M**	**Mo:Ch.2** **25:75**	5.71 ± 0.37 ^cde^	6.57 ± 0.20 ^bc^	3.57 ± 0.20 ^ab^	3.43 ± 0.20 ^cd^	3.71 ± 0.18 ^a^	4.96 ± 0.37 ^cde^	5.56 ± 0.17 ^cde^	5.52 ± 0.17 ^cde^
**PC_3__C**		5.43 ± 0.29 ^bcd^	5.29 ± 0.29 ^ef^	3.43 ± 0.20 ^ab^	2.86 ± 0.26 ^de^	3.57 ± 0.20 ^ab^	3.56 ± 0.29 ^bcd^	4.96 ± 0.29 ^bcd^	4.96 ± 0.29 ^bcd^
**PC_4__M**	**Mo:Ch.4** **75:25**	5.43 ± 0.44 ^def^	6.86 ± 0.26 ^b^	3.29 ± 0.18 ^abc^	3.71 ± 0.18 ^c^	3.43 ± 0.20 ^abc^	5.17 ± 0.44 ^def^	5.90 ± 0.14 ^def^	6.10 ± 0.14 ^def^
**PC_4__C**		5.00 ± 0.37 ^cde^	5.43 ± 0.30 ^ef^	3.14 ± 0.14 ^bcd^	3.57 ± 0.20 ^c^	3.57 ± 0.20 ^ab^	4.10 ± 0.37 ^cde^	5.17 ± 0.17 ^cde^	5.17 ± 0.17 ^cde^
**PC_5__M**	**Mo:Ch.4** **50:50**	6.57 ± 0.26 ^bc^	7.00 ± 0.31 ^ab^	3.57 ± 0.20 ^ab^	4.00 ± 0.22 ^bc^	3.71 ± 0.29 ^a^	5.86 ± 0.26 ^bc^	5.96 ± 0.26 ^bc^	6.96 ± 0.26 ^bc^
**PC_5__C**		6.14 ± 0.30 ^b^	5.86 ± 0.26 ^cde^	3.57 ± 0.20 ^ab^	3.86 ± 0.26 ^bc^	3.86 ± 0.26 ^a^	4.97 ± 0.30 ^b^	5.52 ± 0.30 ^b^	5.52 ± 0.30 ^b^
**PC_6__M**	**Mo:Ch.4** **25:75**	8.57 ± 0.31 ^a^	7.71 ± 0.29 ^a^	3.86 ± 0.14 ^a^	5.00 ± 0.22 ^a^	3.57 ± 0.37 ^ab^	7.10 ± 0.31 ^a^	6.20 ± 0.21 ^a^	6.30 ± 0.21 ^a^
**PC_6__C**		8.00 ± 0.30 ^a^	6.29 ± 0.18 ^bcd^	3.71 ± 0.29 ^ab^	4.43 ± 0.20 ^ab^	3.57 ± 0.37 ^ab^	5.90 ± 0.30 ^a^	5.90 ± 0.10 ^a^	5.90 ± 0.10 ^a^

Means within a column with different superscripts differ (*p* < 0.05), M = Microwave, C = Conventional Oven; Mo = Mozzarella cheese, Ch. = Cheddar cheese, Ch.2 = 2 months ripened Cheddar cheese, Ch.4 = 4 months ripened Cheddar cheese; values given are the means of the three replicates

**Table 4 foods-09-00214-t004:** Effect of proportion of cheeses, ripening time (months) of Cheddar cheese, and cooking methods on the texture descriptors of Pizza cheese.

Treatments	Cheese Proportions	Texture Descriptors (Mean ± SE)			
		Stringiness	Stretchability	Firmness	Tooth Pull
**PC_0__M**	**Mo:Ch** **100:0**	5.57 ± 0.20 ^gh^	8.43 ± 0.20 ^a^	8.00 ± 0.31 ^ab^	8.00 ± 0.31 ^a^
**PC_0__C**		5.43 ± 0.43 ^h^	8.00 ± 0.31 ^ab^	8.43 ± 0.20 ^a^	7.57 ± 0.20 ^a^
**PC_1__M**	**Mo:Ch.2** **75:25**	5.86 ± 0.34 ^fgh^	7.43 ± 0.37 ^bc^	7.29 ± 0.29 ^bc^	6.57 ± 0.20 ^b^
**PC_1__C**		6.43 ± 0.20 ^efg^	6.86 ± 0.26 ^cde^	7.57 ± 0.20 ^ab^	6.14 ± 0.14 ^bc^
**PC_2__M**	**Mo:Ch.2** **50:50**	7.00 ± 0.22 ^cde^	7.00 ± 0.31 ^cd^	5.57 ± 0.20 ^ef^	5.43 ± 0.20 ^de^
**PC_2__C**		6.57 ± 0.30 ^def^	6.43 ± 0.20 ^de^	6.57 ± 0.37 ^cd^	5.29 ± 0.29 ^de^
**PC_3__M**	**Mo:Ch.2** **25:75**	7.43 ± 0.20 ^bcd^	6.57 ± 0.30 ^de^	5.00 ± 0.31 ^fg^	4.14 ± 0.14 ^h^
**PC_3__C**		7.71 ± 0.18 ^abc^	6.29 ± 0.36 ^de^	5.71 ± 0.18 ^def^	4.14 ± 0.26 ^h^
**PC_4__M**	**Mo:Ch.4** **75:25**	7.43 ± 0.30 ^bcd^	6.29 ± 0.36 ^de^	6.14 ± 0.34 ^de^	5.71 ± 0.18 ^cd^
**PC_4__C**		7.86 ± 0.40 ^abc^	6.29 ± 0.29 ^de^	6.57 ± 0.37 ^cd^	5.43 ± 0.20 ^de^
**PC_5__M**	**Mo:Ch.4** **50:50**	7.57 ± 0.57 ^bc^	6.71 ± 0.29 ^cde^	4.57 ± 0.37 ^g^	4.86 ± 0.14 ^efg^
**PC_5__C**		7.86 ± 0.40 ^abc^	6.14 ± 0.26 ^e^	5.14 ± 0.40 ^fg^	5.00 ± 0.44 ^ef^
**PC_6__M**	**Mo:Ch.4** **25:75**	8.57 ± 0.20 ^a^	5.29 ± 0.29 ^f^	4.43 ± 0.30 ^g^	4.43 ± 0.20 ^fgh^
**PC_6__C**		8.29 ± 0.18 ^ab^	4.86 ± 0.34 ^f^	4.29 ± 0.36 ^g^	4.29 ± 0.18 ^gh^

Means within a column with different superscripts differ (*p* < 0.05); M = Microwave cooking, C = Conventional Oven cooking; Mo = Mozzarella cheese, Ch. = Cheddar cheese, Ch.2 = 2 months ripened Cheddar cheese, Ch.4 = 4 months ripened Cheddar cheese; values are the means of the three replicates.

**Table 5 foods-09-00214-t005:** Effect of proportion of cheeses, ripening time (months) of Cheddar cheese and cooking methods on the appearance descriptors of Pizza cheese.

Treatments	Cheese Proportions	Appearance Descriptors (Mean ± SE)			
		Meltability	Oiliness	Edge Browning	Surface Rupture
**PC_0__M**	**Mo:Ch.** **100:0**	6.71 ± 0.18 ^efg^	6.57 ± 0.30 ^f^	4.71 ± 0.29 ^g^	6.00 ± 0.22 ^bc^
**PC_0__C**		6.43 ± 0.20 ^g^	6.43 ± 0.30 ^f^	5.00 ± 0.31 ^g^	8.00 ± 0.22 ^a^
**PC_1__M**	**Mo:Ch.2 75:25**	7.14 ± 0.14 ^def^	6.86 ± 0.26 ^ef^	5.43 ± 0.30 ^fg^	5.71 ± 0.29 ^c^
**PC_1__C**		6.57 ± 0.20 ^fg^	6.43 ± 0.20 ^f^	6.00 ± 0.22 ^ef^	6.43 ± 0.20 ^b^
**PC_2__M**	**Mo:Ch.2** **50:50**	7.71 ± 0.29 ^bcd^	7.43 ± 0.20 ^de^	6.71 ± 0.29 ^de^	3.29 ± 0.18 ^e^
**PC_2__C**		7.14 ± 0.26 ^def^	7.00 ± 0.31 ^ef^	7.00 ± 0.31 ^d^	4.29 ± 0.18 ^d^
**PC_3__M**	**Mo:Ch.2** **25:75**	8.29 ± 0.18 ^ab^	8.14 ± 0.26 ^bc^	6.71 ± 0.36 ^de^	1.86 ± 0.26 ^fg^
**PC_3__C**		8.14 ± 0.14 ^abc^	7.71 ± 0.29 ^cd^	7.43 ± 0.30 ^cd^	3.43 ± 0.20 ^e^
**PC_4__M**	**Mo:Ch.4** **75:25**	7.71 ± 0.29 ^bcd^	8.43 ± 0.20 ^ab^	7.00 ± 0.31 ^d^	4.57 ± 0.20 ^d^
**PC_4__C**		7.29 ± 0.36 ^de^	7.71 ± 0.18 ^cd^	7.86 ± 0.26 ^bc^	5.71 ± 0.29 ^c^
**PC_5__M**	**Mo:Ch.4** **50:50**	8.14 ± 0.26 ^abc^	8.86 ± 0.14 ^a^	7.86 ± 0.14 ^bc^	2.00 ± 0.22 ^f^
**PC_5__C**		7.57 ± 0.30 ^cd^	8.29 ± 0.29 ^abc^	8.29 ± 0.29 ^ab^	3.29 ± 0.18 ^e^
**PC_6__M**	**Mo:Ch.4** **25:75**	8.71 ± 0.18 ^a^	8.71 ± 0.18 ^ab^	8.57 ± 0.20 ^ab^	1.29 ± 0.18 ^g^
**PC_6__C**		8.29 ± 0.18 ^ab^	8.57 ± 0.20 ^ab^	8.71 ± 0.18 ^a^	2.29 ± 0.18 ^f^

Means within a column with different superscripts differ (*p* < 0.05); M= Microwave cooking, C = Conventional oven cooking; Mo = Mozzarella cheese, Ch. = Cheddar cheese, Ch.2 = 2 months ripened Cheddar cheese, Ch.4 = 4 months ripened Cheddar cheese; values given are the means of the three replicates.

**Table 6 foods-09-00214-t006:** Result of PCA on the descriptive sensory data of Pizza cheese showing the loadings of each variable on the first two principal components.

(**a**)										
**Flavor Descriptors**	**Cheddar**	**Acidic**	**Rancid**	**Bitter**	**Salty**	**Sweet**	**Moldy**	**Creamy**	**Variance**	
**PC1**	−0.240	−0.244	−0.265	−0.267	−0.219	0.321	−0.143	0.341	78.8%	
**PC2**	−0.169	0.314	0.166	0.029	0.218	0.243	−0.162	−0.165	10.9%	
(**b**)										
**Sensory Descriptors**	**Texture Descriptors**				**Appearance Descriptors**					**Variance**
	**Stringiness**	**Stretch-Ability**	**Firmness**	**Tooth Pull**	**Melt-Ability**	**Oiliness**	**Edge Browning**		**Surface Rupture**	
**PC1**	−0.268	0.256	0.273	0.263	−0.260	−0.258	−0.265		0.255	78.8%
**PC2**	−0.079	0.226	−0.103	0.097	0.222	0.144	−0.189		−0.222	10.9%

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
