# Peer review of "Descriptive Sensory Analysis of Pizza Cheese Made from Mozzarella and Semi-Ripened Cheddar Cheese Under Microwave and Conventional Cooking"

_foods, 2020, doi:10.3390/foods9020214_

Round 1
Reviewer 1 Report
The article submitted to FOODS entitled “Descriptive sensory behavior of pizza cheeses made from mozzarella and semi-2 ripened cheddar cheese under microwave and conventional cooking” aims to optimize the best blend of Mozzarella and Cheddar, with different ripening times) to be used on Pizza. Additionally, the authors compared the effect of cooking by microwaves or conventional oven.
I found the objective interesting, but I have several doubts about the adequacy of the methodology used, that are listed below.
There is a problem with the references, once there are numbers in the text that do not have correspondence in the list of references, and other references seem to do not correspond to the number in the text.
In the material and methods section, there is a part o text that is repeated.
Am I not a native English speaker. Having that in consideration, I found the language adequate for an international journal but needs language corrections.
Comment #1. The core of this article is based on sensory analysis. According to my understanding of the article, some aspects need to be clarified:
Descriptive analysis is an analytical method. It is expected to be made by a trained and selected panel, and the evaluation is expected to be qualitative. Hedonic tests are expected to be made with consumers, with more individuals, that do not need to be particularly selected or trained. In this article, if I understood correctly, 12 semi trained judges were used, but the aim was to know if they like the character or not (hedonic scale) If there is a mistake, and the scale was quantitative, it is legit to use the 12 judges – how were they semi-trained (a node on that procedure should be included) If the approach is hedonic, it is not a descriptive test, it is a hedonic test for the several characteristics If the sensory analysis was an accessory approach in an article, having the aim in another analytical tool – chemical or microbiological – we could be less exigent in the adequacy in the sensory methodology. In the present work, the core is sensory analysis, thus, the correctness of the methodology must be guaranteed
For details please check the guidelines on https://www.sensorysociety.org/knowledge/Pages/Sensory-Data-Publications.aspx
QUOTING THE SENSORY SOCIETY
“Determining the Test
The test objective should help define the test and parameters (samples, respondents, location, etc.) that is chosen. When more than one objective is present, it may be necessary to conduct more than one type of sensory test.
The determination of liking or acceptance requires the use of a consumer test; trained descriptive panelists should never be used for this purpose. Determining intensity information about attributes can be done either by trained descriptive panelists or by consumers depending on the level of detail that is needed. Trained panelists typically provide more detailed information than untrained consumers. To determine if samples are different overall from each other, discrimination tests often are used. When using discrimination tests, such as triangle or duo-trio tests, it is important to conduct them with panelists who have been screened to ensure they can find differences in similar product categories. Regardless of the test structure that is chosen, the test must be statistically powerful to determine when key differences are or are not present.END OF QUOTING
Comment #2. Still related to comment #1m the paragraph in line 43-52 raised few doubts. What do the authors mean with “but generic descriptive analysis used a combination of descriptive analysis to meet specific objectives”?
In line 49 – “identify and quantify sensory aspects of a food” I totally agree with that. That is why my question on comment #1 on the methodology used in the present work – the phrase used after (line 66) "consumers liking" is typically used for consumer tests, that should not be done with 12 assessors…
Line 59 – "ultimate pizza cheese – please consider rephrasing – optimization of pizza …
Line 62-64 – The example of juices is out the context – juices are a very different food to be used in this context. Additionally, I believe that in that context it is used as a pasteurization method, thus its pertinence here is doubtful.
Line 77-83 – I did nor understood which references support the preparation of the cheese. How was it amalgamated? Was it melted and combined or was it combined after shredding? Once it is the core of the article details on that preparation should be provided,.
In this section, it is not necessary to explain how will it be cooked, once it is referred after in the sensory analysis
Line 85-86 – Why one reference for each component, if at the end there is a reference for all (I believe that the numbers in the text do not correspond to the references in the reference list)
Line 90 – the power od microwave cooking should be included.
Line 93-94 – How was the selection made?
From line 106 – repeated
A heading on statistical analysis should be included, with details of the analysis
Tables – it is quite difficult to read the tables with the numbers of the blends; maybe if authors used the proportion of each in the heading it would be easier to read.
Table 2 (table 1 cannot find it!) What is different between Mozzarella and PC0 – if it is 100% mozzarella.
In this discussion, the composition is highly dependent on the proportion of each cheese,: mozzarella with more water and cheddar with less water – the other parameters are mainly a consequence of that proportion, rather than the arguments presented in lines 135-138.
I stop here the detailed revision, once, in my opinion, the article needs serious clarification on the aspects referred to in comment#1 and comment #2 to be further analysed.
Reviewer 2 Report
Table 1 missingLines 23 and 29: it would be better to always use capital letters to indicate sensory attributes.
Row 79 and 292 and table 6: identification of the samples with the abbreviation PC causes confusion with the processing of data by the PCA
The procedure for identifying and selecting descriptors is not indicated
Line 87 and Line 117: “Descriptive sensory evaluationof Pizza cheese” is repeated
Data processing is interesting but there is no adequate description of how the data is collected.
Lines 95: use capital letters to indicate sensory attributes
The description of the test procedure with the sensory judges is missing (sample preparation, administration, sensory laboratory, replicas, procedures, etc ...)
No official procedure is applied (ISO 11035; UNI EN ISO 13299; ISO 11036; etc.)
Reviewer 3 Report
Extensive revision of the manuscript is proposed. The manuscript should be revised by an english native speaker. Sensory analysis methodology used must be clearly explained.

Round 2
Reviewer 1 Report
The revision made to the paper addressed the concerns I had in the first revision. The authors made the changes necessary and clarified the aspects that I had doubts about.
Just a note. I do not see the need for having the vocabulary used in the sensory analysis in CAPS in the abstract.
Reviewer 2 Report
Thanks for the revisions and the satisfactory answer provided
Reviewer 3 Report
I strongly recommend the revision of the manuscript by an english native speaker. Please find revision sugestions in the attached document.
